# Chemico-Pharmacological Screening of the Methanol Extract of *Gynura nepalensis* D.C. Deciphered Promising Antioxidant and Hepatoprotective Potentials: Evidenced from *in vitro*, *in vivo*, and Computer-Aided Studies

**DOI:** 10.3390/molecules27113474

**Published:** 2022-05-27

**Authors:** Nishan Chakrabarty, Hea-Jong Chung, Rashedul Alam, Nazim Uddin Emon, Safaet Alam, Mohammed Fazlul Kabir, Md. Minarul Islam, Seong-Tshool Hong, Tapas Sarkar, Md. Moklesur Rahman Sarker, Mohammad Manjur Rahman

**Affiliations:** 1Department of Pharmacy, Jagannath University, Dhaka 1100, Bangladesh; nishaniiuc@gmail.com; 2Gwanju Center, Korea Basic Science Institute, Gwanju 61715, Korea; 3Department of Pharmacology, Medical School, Jeonbuk National University, Jeonju 54896, Korea; rashedcmu@gmail.com; 4Department of Pharmacy, Faculty of Science and Engineering, International Islamic University Chittagong, Chattogram 4318, Bangladesh; nazim7emon@gmail.com; 5Pharmaceutical Sciences Research Division, Bangladesh Council of Scientific and Industrial Research Laboratories (BCSIR), Dr. Qudrat-I-Khuda Road, Dhanmondi, Dhaka 1205, Bangladesh; 6Department of Biology, Georgia State University, Atlanta, GA 30303, USA; kabirrasel07@gmail.com; 7Department of Biomedical Sciences, Institute for Medical Science, Jeonbuk National University Medical School, Jeonju 54907, Korea; mislambcmb@gmail.com (M.M.I.); seonghong@jbnu.ac.kr (S.-T.H.); 8Department of Pharmacy, Gono Bishwabidyalay, Dhaka 1344, Bangladesh; taposh12.bd@gmail.com; 9Department of Pharmacy, State University of Bangladesh, Dhaka 1205, Bangladesh; prof.moklesur@sub.edu.bd or; 10Pharmacology and Toxicology Research Division, Health Med Science Research Limited, 3/1 Block F, Lalmatia, Mohammadpur, Dhaka 1207, Bangladesh; 11Department of Biochemistry and Molecular Biology, Medical University of South Carolina, Charleston, SC 29425, USA; 12Department of Biological Sciences, St. John’s University, Queens, NY 11439, USA

**Keywords:** *Gynura nepalensis*, hepatoprotective, antioxidant, bioactive compound, phytomedicine, binding affinity, complementary and alternative medicine

## Abstract

*Gynura nepalensis* D.C. (family: Asteraceae) has abundant uses in the alternative medicinal practice, and this species is commonly used in the treatment of diabetes, rheumatism, cuts or wounds, asthma, kidney stones, cough, urinary tract bleeding, gall bladder stones, hepatitis, diarrhea, hemorrhoids, constipation, vomiting, fertility problems, blood poisoning, septicemia, skin allergy, indigestion, high cholesterol levels, and so on. This study aims to investigate the hepatoprotective and antioxidant potential of the methanol extract of the *Gynura nepalensis* D.C. (GNME) along with chemical profiling with phytochemical screening. Moreover, prospective phytocompounds have been screened virtually to present the binding affinity of the bioactive components to the hepatic and oxidative receptors. In the hepatoprotective study, alanine transaminase (ALT), aspartate aminotransferase (AST), alkaline phosphatase (ALP), total protein (TP), and lipid peroxidation (LP) and total bilirubin (TB) have been assessed, and in the antioxidant study, the DPPH free radical scavenging, total antioxidant flavonoid, and phenolic contents were determined. Moreover, the molecular binding affinity of the bioactive component of the plant has been analyzed using PyRx AutoDock Vina, Chimera, and Discovery Studio software. The plant extract showed dose-dependent hepatoprotective potential (*p* < 0.05, 0.01, 0.001) as well as strong antioxidant properties. Moreover, hepatoprotective and antioxidant molecular docking studies revealed a result varying from −2.90 kcal/mol to −10.1 kcal/mol. 4,5-dicaffeoylquinic acid and chlorogenic acid revealed the highest binding affinity among the selected molecules. However, the plant showed portent antioxidant and hepatoprotective properties in the in vitro, in vivo, and in silico models, and it is presumed that the hepatoprotective properties of the plant extract have occurred due to the presence of the vast bioactive chemical compounds as well as their antioxidant properties. Therefore, advanced studies are recommended to elucidate the pharmacological properties of the plant extracts.

## 1. Introduction

The liver plays a significant role in energy metabolism and xenobiotic biotransformation. As a consequence, prolonged exposure to harmful xenobiotics is probable to occur in liver damage, which may lead to cancer, cirrhosis, and acute liver failure [1]. Because of its multi-biological activities in protein, lipid, and carbohydrate metabolism, this is one of the most essential body organs. Acute and chronic liver illnesses have become a worldwide problem, and medical therapies for them are sometimes difficult to obtain and may be ineffective [2]. Liver diseases have the possibility to be fatal, posing a danger to global public health. As a consequence, interest in complementary and alternative treatments for the treatment of hepatic disorders has increased significantly. When these drugs (e.g., Bicyclol, Calmangafodipir, Cytisin amidophosphate, Fomepizole, Livina, Magnesium isoglycyrrhizinate, S-adenosylmethionine, Picroliv, Radix paeoniae rubra) [3] are used in clinical trials, generating therapeutically useful molecules from natural sources may help to reduce the risk of adverse events [4]. However, despite the fact that paracetamol is widely used as an analgesic and antipyretic, it may cause liver and kidney damage at large concentrations [5]. Paracetamol is converted to the dangerous metabolite NAPQI (N-acetyl-p-benzoquinoneimine) by cytochrome P450 enzymes, which deteriorate glutathione levels (GSH) [6], and numerous cellular proteins, notably mitochondrial proteins, are bound to NAPQI, leading to mitochondrial oxidative stress, cell death, and necrosis [7]. Living systems generate free radicals as a byproduct of their normal metabolic processes. There are three most prevalent reactive oxygen species (ROS): the hydroxyl radical (OH-), the superoxide anion (O_2_-), and hydrogen peroxide (H_2_O_2_). Secondary metabolites from plants, such as phenolic chemicals, may be used as antioxidants, whether they are synthetic or natural. However, butylated hydroxyanisole and butylated hydroxytoluene are synthetic antioxidants that are found to be carcinogenic [8,9]. Free radical reactions are well known to have a role in disease pathogenesis. Although these reactions are necessary for proper metabolism, they may also be damaging to one’s health, leading to the development of illnesses such as diabetes, neurodegenerative disorders and immunosuppression [10]. Paracetamol-induced hepatotoxicity is characterized by oxidative stress. Hepatoprotective effects of antioxidant phytochemicals against paracetamol-induced acute liver injury in rats have been reported previously [9]. Moreover, docking studies are often used to anticipate interactions between molecules and proteins which is a handy tool to acquire biological activity information from natural sources using molecular docking research. In addition, it gives more information on the interactions and the mechanisms by which certain proteins’ binding sites are expected to operate [11].

Hepatopathy is one of the most serious conditions for which herbal medicines are being used in numerous nations. Medicines that are presently on the market were first accomplished through the traditional knowledge about medicinal plants [9,12]. As the expense and toxicity of modern pharmaceuticals continue to rise, people are turning to plant-based medicinal products in the hope of being relieved from the diseases [13]. Consequently, many people have shifted their focus to medicinal plants since they are less expensive and easier to obtain [14]. An increasing number of people across the globe are also turning to natural derivatives for liver diseases because of the widespread belief that they are safer and have fewer adverse events [15]. It is possible to obtain phytochemicals from the bark of the plant as well as the fruit, seeds, roots, leaves, and roots. Research on plant species and their derivatives is being carried out in several laboratories throughout the globe to uncover bioactive chemicals to treat diseases. Research on herbal plants must be expanded in order to discover novel chemical compounds with pharmacological effects [16]. Many phytochemical-based formulations are available worldwide to treat liver disease. A simple and exact herbal remedy for the treatment of liver disorders is still a fascinating topic of study [17]. The search for plants with hepatoprotective and antioxidant properties was thus a key priority for this research project.

The Asteraceae family comprises *Gynura nepalensis* D.C., a medicinal herb with several therapeutic uses. As far west as Kashmir, *G. nepalensis* may be found in India, Himalayas, Bangladesh, and China, as well as Nepal, Thailand, and the Philippines. Perennial herb with a long, gray pubescence and fibrousness is a characteristic of roots. Decumbent stems with a corymbosely branched upper portion are 30-45 cm in height, woody at the base with a diameter of around 10 mm. *G. nepalensis* contains hydroxy-β-ionone, chlorogenic acid (3-O-caffeoylquinic acid), 3-p-coumaroylquinic acid, 3,4-dicaffeoylquinic acid, 9′-O-ethyl-dehydrodiconiferyl alcohol, 4,5-dicaffeoylquinic acid (isochlorogenic acid C), ethyl caffeate, loliolide, 4,5-dicaffeoylquinic acid methyl ester, dibutyl phthalate, 2-(1H-indol-3-yl)-2-oxoacetamide, (+)-medioresinol, 7S,8R-9′-O-ethyl-dehydrodiconiferyl-9-acetate, 1H-indole-3-carbaldehyde, propiconazole, boscialin, 3,6-trans-3-hydroxy-α-ionone, 4-hydroxy-4,7-dimethyl-1-tetralone, dibenzothiophen-5-oxide, trans-4,5-dihydroxycorocalane [18]. A wide range of human ailments can be treated with *G. nepalensis* which is an ethnomedicine to treat hepatitis, coughing and asthma, diabetes, indigestion cuts and wounds, as well as kidney and gall bladder stones, hemorrhoids, and urinary tract bleeding. It also treats blood poisoning and fertility issues, as well as septicemia, rheumatism, and skin allergies. It is also used to treat high cholesterol and low blood pressure [19,20,21,22,23]. Despite the fact that this plant has been used to treat a number of diseases, no study on its antioxidant and hepatoprotective properties has been conducted. Hence, we aimed to explore the heptatoprotective and antioxidant potentials of GNME by employing in vitro and in vivo studies. We also aimed to perform the molecular docking of phytoconstituents to anticipate the binding affinity to exert the antioxidant and hepatoprotective activities of GNME.

## 2. Results

### 2.1. Qualitative Screening of Phytochemicals

Plant metabolites were determined by conducting a qualitative phytochemical screening. Table 1 shows the phytochemical screening findings for GNME. Alkaloids, Carbohydrates, Saponins, Tannins, Condensed Tannin, Terpenoids, Chlorogenic acid, Steroidal Glycosides, Anthocyanin, Flavonoids, Flavones, Phenols, Coumarins, and Nitrogenous compounds were found to be present, whereas Quercetin, Triterpene, and Coumarin were absent in the primary screening.

### 2.2. Acute Toxicity Assay

The oral acute toxicity study of GNME was carried out following OECD guideline fixed-dose procedure (OECD protocol no. 420) as mentioned in Section 4. No case of mortality was observed during the 14 days of treatment with a limit dose of 3000 mg/kg BW of GNME. All treated animals could tolerate the GNME doses, and there was no statistically significant difference in body weight between the treated and untreated groups. The animals did not exhibit any abnormalities or major behavioral changes. The hair, face, eyes, and nose were unaffected in an acute oral toxicity test. For example, there were no signs of tremors, visions, salivation, or diarrhea. Behaviors, such as regressive gestures, posture, and emotional state, were also in line. Weighing remained steady in both control and treatment groups. Each animal was fed and watered in the same way. Based on the toxicity study results, it was found that the GNME was apparently safe to administer orally up to the single dose of GNME 3000 mg/kg/p.o. Therefore, the doses of GNME (100, 200, and 400 mg/kg) used in our investigation were found to be safe [15].

### 2.3. Hepatoprotective Effects of GNME

Paracetamol has enhanced the levels of SGPT, SGOT, bilirubin (both total and direct bilirubin levels), and alkaline phosphatase level (ALP). Treatment with silymarin and 200 mg/kg and 400 mg/kg of GNME has significantly brought down the elevated levels of SGPT, SGOT, ALP, and bilirubin. In the present study, GNME exhibited a dose-dependent and statistically significant effect. Administration of GNME 100 mg/kg and 200 mg/kg suppressed AST by 20.04% (*p* < 0.001) and 38.88% (*p* < 0.001) respectively and ALT by 48.37% (*p* < 0.001) and 67.53% (*p* < 0.001) respectively when compared to negative control. GNME 400 mg/kg dose showed an even better effect than standard Silymarin, where GNME 400 mg/kg inhibited AST and ALT by 61.12% and 78.68%, respectively, and Silymarin reduced AST and ALT by 41.63% and 72.14%, respectively while comparing with negative control. A dose-dependent reduction in liver weight was seen after the administration of GNME (100 to 400 mg/kg; b.w; p.o), as shown in Table 2. Moreover, the effects of GNME on hepatoprotectivity have been described briefly in Table 3.

### 2.4. Antioxidant Effects of GNME

#### 2.4.1. DPPH Radical Scavenging Assays

One of the most often used and reliable methods for measuring free radical scavenging or antioxidant capacity using the DPPH method. In the DPPH radical scavenging assay, the GNME showed dose-dependent scavenging equivalent to the baseline antioxidant ascorbic acid. The effect of Ascorbic acid and GNME in the DPPH radical scavenging is shown in Table 4 and Table 5 and Figure 1.

#### 2.4.2. Total Antioxidant Capacity

The overall antioxidant activity of GNME was determined by their capacity to reduce the Phosphate/Mo (VI) complex to Phosphate/Mo (V). GNME demonstrated a moderate (106.40 ± 1.19 mg AAE/mg of extract preparation) cumulative antioxidant potential in this analysis. The summary of the antioxidant potential of the plant extract is shown in Table 6 and Figure 1.

#### 2.4.3. Total Flavonoid Contents

The aluminum chloride colorimetric method was used to determine the total flavonoid contents of GNME. The total flavonoid content was calculated using the standard curve of quercetin (y = 0.0101x − 0.007, R^2^ = 0.992) and was expressed as mg quercetin equivalents (QE) per mL of liquid preparation. The total flavonoid contents of the GNME have been illustrated in Table 6 and Figure 2.

#### 2.4.4. Total Phenol Contents

The Folin–Ciocalteu reagent was used to determine the total phenolic content of GNME, which was represented as mg Gallic acid equivalents (GAE) per mL of liquid solution. Test sample total phenolic content was measured by plotting the sample against a Gallic acid reference curve (y = 0.0373x + 0.0681, R^2^ = 0.0971). Total phenolic contents of gallic acid and GNME have been mentioned in Table 6 and Figure 3.

### 2.5. Screening of Phytocompounds for Molecular Binding Affinity

3-P-Coumaroylquinic acid, 4,5-Dicaffeoylquinic acid, Boscialin, Chlorogenic acid, Ethyl caffeate, Dibutyl phthalate, Loliolide, Medioresinol, and Propiconazole are the major bioactive phytochemicals reported from *G. nepalensis* [18] which are exploited to perform computer-aided investigations in this study. The screened molecules showed very strong binding affinities to the Urate oxidase (Uox), Glutathione reductase, Hepatitis C Virus NS3/4A Protease, and Human IgG Fc Domain. In the antioxidant study, Chlorogenic acid yielded the best binding score to the Urate oxidase receptor. The ranking of the affinities are: chlorogenic acid > propiconazole > loliolide > ethyl caffeate > boscialin > medioresinol > dibutyl phthalate > 3-P-Coumaroylquinic acid > 4,5-Dicaffeoylquinic Acid. Again, 4,5-Dicaffeoylquinic acid was bound and scored -10.1 (kcal/mol) when it has interacted with the Glutathione reductase enzyme. The ligand binds to these receptors through a number of amino acid residues, namely: gly62, ser177, lys66, leu338, val370, thr57, ala155, asp331, thr339, ala342. Hepatitis C Virus NS3/4A Protease and Human IgG Fc Domain were used to screen the hepatoprotective activity of the selected molecules. 4,5-Dicaffeoylquinic acid also demonstrated the highest binding score and binding affinity to the Hepatitis C Virus NS3/4A Protease and Human IgG Fc Domain receptors. 4,5-Dicaffeoylquinic acid hit the Hepatitis C Virus NS3/4A Protease in −7.8 Kcal/mol. 4,5-Dicaffeoylquinic binds to the Human IgG Fc Domain by a series of amino acid residues (pro396, ser375, thr393, asp376, pro247). Therefore, the binding scores draw very strong binding affinities to the respective receptors, and the elaborate results are displayed in Figure 4 and Figure 5 and Table 7.

## 3. Discussion

The probable extinction of an estimated 4000 to 10,000 species of medicinal plants on a local, national, regional, or global scale is predicted to have severe consequences for the standard of living, economies, and healthcare [24]. Medicinal plants are the most reliable sources of novel bioactive compounds for the development of new medicines those exert a wide spectrum of pharmacological actions, including neuroprotective, thrombolytic, antidepressants, antioxidants, cytotoxicity, anxiolytics, thrombolytics, and hepatoprotective effects [25]. In continuation, GNME also displayed promising hepatoprotective and antioxidant actions in our experimental models. Compared to normal control rats, the activity of the enzymes AST, ALT, and ALP (hepatic marker enzymes for liver damage), lipid peroxidation, and total bilirubin were significantly enhanced in the extended dose of Paracetamol-treated rat models [6]. Hepatocytes have a high concentration of AST in their mitochondria [6]. Since ALT is more specific to the liver, it is a superior measure for identifying liver damage. The levels of ALP and bilirubin in the blood are also linked to liver cell destruction. The activity of the enzymes ALT, AST, and ALP, as well as the quantity of blood bilirubin, are the most often used biochemical indicators to assess liver impairment [26]. Amino transferases, a class of enzymes that play critical roles in the relationship between carbohydrate and amino acid metabolism, are abundant in the liver. Two enzymes, alanine aminotransferase and aspartate aminotransferase are well known for their capacity to detect liver injury. These indicator enzymes are released into the bloodstream after events in liver injury with hepatic lesions and parenchymal cell necrosis. Elevated plasma levels of alkaline phosphatase, a membrane-bound enzyme, also suggests that the organ’s membranes have been damaged [27]. Although alkaline phosphatases are not specifically liver enzymes, the liver is the primary source of this enzyme. In cholestasis, the amount of this enzyme rises. Cirrhotic liver disease causes hepatotoxicity, which increases bilirubin production [10]. In this study, the paracetamol-treated patients had a reduced protein composition of their livers as well as showed notable changes in liver function parameters ascertained liver damage. Paracetamol hepatotoxicity has been connected to cirrhosis, hepatitis, or liver failure with a number of additional side effects. Overdose of paracetamol, an analgesic medicine, has been linked to significant liver damage in people and animals [28,29]. The major metabolites of paracetamol are glucuronic acid and sulfate conjugation. Multi-drug resistance protein (Mrp2, Mrp3, or both) excrete it into the bile, depending on the conjugated molecule. When used in large amounts, paracetamol (acetaminophen) may cause serious liver damage [3]. Furthermore, when the paracetamol become activated by the hepatic enzyme cytochrome p-45065-66 to form the very reactive metabolite, N-acetyl-P-benzoquinone imine (NAPQI), the hepatotoxicity has been induced, which is linked to the production of hazardous metabolites such as; N-acetyl-P-benzoquinone imine (NAPQI) [30]. After binding covalently to protein cysteine groups, NAPQI produces three (cysteine-S-yl) paracetamol adducts. Hepatocytes are protected when glutathione (GSH) interacts with paracetamol’s reactive metabolite to block covalent attachment to liver proteins. However, overdosing causes the glucuronic acid and sulfate conjugation to become saturated, lowering GSH levels. When NAPQI interacts with hepatic proteins, necrosis develops in the liver [30]. As a consequence, the toxicity of paracetamol is governed by the cytochromes P450’s conversion of the drug to reactive metabolites, but the metabolism can obviously be disrupted by pretreating or cotreating with a phytochemical [31,32]. GNME showed significant hepatoprotective events, and besides the hepatoprotective activities, it also yielded strong antioxidant properties in vitro and in silico research. Total flavonoid and phenol contents, antioxidant capacity, DPPH radical scavenging assay established the evidence of antioxidant properties in the methanol extract of *G. nepalensis*. This plant exhibits significant antioxidant activity, according to the DPPH free radical scavenging test, and the flavonoid content of the extract, which may be responsible for its hepatoprotective properties playing a key role in protecting the liver cells from oxidative stress [33]. Additionally, phytochemicals identified from this plant in previous studies were also considered in this study to establish our study results empirically. Phytochemicals have been reported to exert hepatoprotective actions by reducing the extent of plasma transaminases and ameliorating the histological signs of acute liver damage [34]. β-Sitosterol is a common triterpenoid found in several plant is another hepatoprotective phytochemical, which can regulate liver enzymes levels and other liver markers [35]. Chlorogenic acid, another prospective phytochemical of this plant, can also play a key note to modulate liver enzymes. Loliolide and ethyl caffeate found in *G. nepalensis* are also reported in previous studies as promising antioxidant agents against oxidative damages [36,37]. To recapitulate, presence of several types of bioactive phytochemicals of *G. nepalensis* reported in previous studies along with the current investigation can be the justification and driving force to validate its antioxidant and hepatoprotective potentials. In addition, in silico studies demonstrated that the bioactive constituents of the *G. nepalensis* bind to the Urate oxidase (range: −6.1 to −7.8 kcal/mol) Glutathione reductase (range: −4.8 to −10.1 kcal/mol), Hepatitis C Virus NS3/4A Protease (range: −5.2 to 7.8 kcal/mol) and Human IgG Fc Domain (range: −4.4 to −6.5 kcal/mol) very strongly. Therefore, to conclude, GNME yielded dose-dependent activity in the hepatic degeneration model, and it also showed strong antioxidant properties in the in vitro studies. Moreover, in the molecular docking analysis, it can be said that the selected phytoconstituents have a greater possibility of being a pharmaceutically active agent. However, advanced studies are required to quest the actual mechanism of action and drug safety issues to determine and prepare the pharmaceutically active ingredient from the extracts of *G. nepalensis*.

## 4. Materials and Methods

### 4.1. Drugs and Chemicals

The investigation made use of analytical-grade drugs and chemicals. The drugs and chemicals were bought from authentic agents and drug manufacturers. Paracetamol (Square Pharmaceuticals Ltd., Dhaka, Bangladesh), G-Ketamine IM/IV Injection (Ketamine HCl, USP, 50 mg/mL), (Gonoshasthaya Basic Chemicals Ltd., Tongi, Bangladesh), Heparin (RotexMedica, Trittau, Germany), Bilirubin Liquicolor-Human, Total Protein Liquicolor-Human, ALP Liquicolor-Human, AST Liquicolor-Human, ALT Liquicolor-Human (HUMAN GmbH, Wiesbaden, Germany), Folin–Ciocalteu reagent, Methanol, Sodium Phosphate (Na_3_PO_4_), Ammonium Molybdate, Absolute Ethanol, KCl, FeCl_3_ (Anhydrous) Thiobarbituric acid (TBA), (Merck, Darmstadt, Germany), Sodium carbonate, Concentrated H_2_SO_4_ (98%), Potassium Acetate, Trichloroacetic acid (TCA), HCl (0.25 N), Butylatedhydroxytoluene (BHT), Cupric Chloride (CuCl_2_·2H_2_O), Neocaproin, Ammonium acetate buffer, pH 7.0 (Merck India Limited, Maharashtra, India), Gallic acid (Sigma Chemicals, St. Louis, MO, USA), Aluminum Chloride, Ferric Chloride (Fe_3_Cl), (Fine Chemicals, Mumbai, India), (Merck India Limited, Mumbai, Maharashtra, India), Methanol (Merck, Wiesbaden, Germany), Quercetin, 1,1-diphenyl-2-picrylhydrazyl (DPPH), (Sigma Chemicals, St. Louis, MO, USA), Ascorbic acid (SD Fine Chem., Ltd., Biosar, India), Potassium ferricyanide [K_3_Fe(CN)_6_] (May and Backer, Dagenhan, Romford, UK), Trichloro Acetic acid (Chemical, New Delhi, India).

### 4.2. Biochemical Analysis

The aspartate aminotransferase (AST), alanine transaminase (ALT), total bilirubin, and serum alkaline phosphatase (sALP) were assayed using assay methods mentioned by [38].

### 4.3. Collection of Plant

It was only after a careful review of the scientific literature that the plant *Gynura nepalensis* D.C. was selected for this investigation. The complete plants were collected during the rainy season in 2017 from the university campus of the Jahangirnagar University, Savar, Bangladesh, and identified by a taxonomist from the National Herbarium of Bangladesh, Mirpur, Dhaka, Bangladesh (specimen voucher number: 43415).

### 4.4. Preparation of Extract

The whole plant was washed and cut into small pieces and shade-dried. Finally, a hot air oven was used to dry the material so that it could be ground to a fine powder. A high-capacity grinding mill was used to crush the plant into a coarse powder, which was then placed in airtight containers with the requisite marks for identification and maintained in a cold, dark, and dry area for the post-analysis. A Soxhlet apparatus was used to extract the powdered plant material (430 gm) using absolute methanol. Colorless liquid siphoning cycles in the Soxhlet device confirmed that the plant components had been extracted to the point where they were depleted of their constituents. Afterward, it was passed through a Whatman number 1 filter paper to remove any remaining impurities. A rotatory evaporator was used to dry the filtrate and produce a gummy concentration of the crude extracts. The filtrate was obtained by using a rotatory evaporator to have a gummy concentrate of the crude extract. The weight of the yielded extracts was 17 g. The extracts were kept in a suitable container with proper labeling and stored in a cold and dry place.

### 4.5. Animals

Wistar rats of both sexes weighing 180–200 g were supplied by Venom Research Centre (VRC), Chittagong. Selected animals were used in pharmacological and toxicological research. The normal 12 h day/night cycle was maintained at the Animal House of the Department of Pharmacy, International Islamic University Chittagong, where they were confined in cages with a well-ventilated room. Rodent pellet feed was supplied to the rats on a regular basis, and water was provided ad libitum. Cages and water bowls were cleaned and refilled every day. The rats were allowed two weeks to adjust to their new settings in the animal laboratory. Prior to conducting animal studies, the Research Ethics Committee of the Department of Pharmacy, International Islamic University Chittagong, Chittagong 4318, Bangladesh, provided its consent under the approval number 140/41/22/02/2018.

### 4.6. Phytochemical Analysis

#### Qualitative Phytochemical Screening

The existence of the Alkaloids, Carbohydrates, Saponins, Tannins, Condensed Tannin, Terpenoids, Chlorogenic acid, Steroidal Glycosides, Anthocyanin, Flavonoids, Flavones, Phenols, Coumarins, and Nitrogenous compounds in the methanol extracts of *G. nepalensis* have been screened by the established protocols mentioned in previous studies [24].

### 4.7. In vivo Analysis

#### 4.7.1. Acute Toxicity Test

The acute toxicity study was conducted following OECD (organization for economic cooperation and development) guidelines by the fixed-dose procedure [39]. A single dose of GNME 300, 2000, and 3000 mg/kg/p.o. were administered. The animals were observed individually during the first 30 min after dosing, and special attention was given during the first 4 h, then observed periodically during the first 24 h to see any toxic effects in the animals. During the entire period of observation for 14 days, the animals were observed and monitored for any changes in behavior, body weight, urination, food intake, water intake, respiration, convulsions, tremor, temperature, and constipation, changes in eye and skin colors and mortality of the animals [40].

#### 4.7.2. Assessment of Paracetamol-Induced Hepatoprotective Activity

Sreedevi et al. [41] used a paracetamol-induced liver damage model to incite hepatotoxicity in their experiments, and the method is also mentioned by Hossain et al. [42]. For the experiment, the 30 rats were split up into the following six groups: to serve as a baseline for the other groups, we used group I animals, who received distilled water at a dose of 10 mL/kg body weight (p.o.) per day for 10 days as a control. Group II was given paracetamol (2 gm/kg; p.o.) from the 6th day of the trial through the 10th day of the experiment. Silymarin (100 mg/kg b.w.; p.o. was administered for 10 days) was given to group III. Group IV got GNME at a dosage of 100 mg/kg of body weight (p.o.). In group V, the GNME was provided at 200 mg/kg b.w. for 10 days, whereas in group VI, the GNME was administered at 400 mg/kg; b.w. As soon as the trial began and continued until the 10th day, subjects in groups II-VI were given paracetamol (2 gm/kg).

#### 4.7.3. Preparation of the Samples for Biochemical Studies

Blood was obtained from the animal’s post vena cava and put in tubes containing heparin shortly after drawing it. The serum in blood samples was separated for biochemical examination by spinning them at 3000 rpm for 10 min. Organs such as the liver, heart, kidney, and brain were blotted, dried, and weighed [43].

#### 4.7.4. Preparation of Liver Homogenate

To create 10% *w/v*, we chopped and homogenized 0.5 g of wet liver tissue in 0.1 N Tris-HCl (pH 7) and 0.15 M KCl. The tissue was centrifuged for 10 min at 3000 rpm. Afterward, the top layer of the separation was collected and employed for further investigation (lipid peroxidation and protein determination assay) [44,45].

#### 4.7.5. Blood Sample Collection

At the end of the experiment, the animals were sacrificed via ether anesthesia, and blood was taken without the use of an anticoagulant for serum preparation. Blood was obtained using retro-orbital puncture and allowed to stand for 10 min before being centrifuged at 2000 rpm for an additional 10 min. The serum was obtained using a micropipette, and the enzymes AST, ALT, ALP, total protein, and total bilirubin were determined.

#### 4.7.6. Assessment of Liver Functions

Analyses of biochemical parameters such as Alanine transaminase (ALT), aspartate aminotransferase (AST), alkaline phosphatase (ALP), total protein (TP), lipid peroxidation (LP), and total bilirubin (TB) have been carried out using kits in the Medinova Diagnostic Center’s DimensionRXL (Max)/vittros-250 autoanalyzer. The protocols were mentioned in Cieslak et al., [46,47].

#### 4.7.7. Observation of Liver Weight

Toxins are typically harmful to the target organ when rats are exposed to dangerous substances. As a consequence, the wounded organ(s) will have an altered weight, either increasing or decreasing. Paracetamol increased liver weight in comparison to the Control group. Silymarin (STD) reduced the rat’s liver weight when compared to the control group.

#### 4.7.8. Determination of ALT, AST, ALP, TP, TB, LP

The Expert Panel of the International Federation for Clinical Chemistry (IFCC) recommends this approach for determining activity levels of the ALT and AST enzymes [48,49]. Alkaline phosphatase of the rat serum was determined by using the “Optimized standard Colorimetric method (Techinico Ames, RA-50, Chemistry Analyzer)” according to the recommendations of the German Clinical Chemistry Association (Human Gesellschaft for Biochemica and Diagnostica GmbH, Wiesbaden, Germany) [50]. Cupric ions react with protein in an alkaline solution to form a purple complex. The absorbance of this complex is proportional to the protein concentration in the sample [51]. Total Bilirubin (TB) and Lipid Peroxidation have been determined by following the method developed by Pearlman et al. [52] and Grespan et al. [53], respectively.

### 4.8. In vitro Antioxidant Analysis

#### 4.8.1. DPPH Free Radical Scavenging Assay

The stock solution was prepared by following the method used by Alam et, al. [40]. The stock solution was serially diluted to achieve the concentrations of 400 µg/mL, 200 µg/mL, 100 µg/mL, 50 µg/mL, 25 µg/mL, and 12.5 µg/mL. Each test tube contains 1 mL of each concentration and is properly marked 2 mL of 0.004% DPPH solution in the solvent is added to each test tube to make the final volume 3 mL (caution: DPPH is light-sensitive, so making the solution and adding it to the test tubes should be performed in minimum light exposure). Incubate the mixture at room temperature for 30 min in a dark place. Then the absorbance is measured at 517 nm against dilute extract solution in the solvent [40].

#### 4.8.2. Determination of Total Antioxidant Capacity

Different test tubes were filled with 300 mL of the plant extracts or standards at various concentrations, followed by the addition of 3 mL of the reagent solution. The reaction in the test tubes was completed after 90 min of incubation at 95 °C. A spectrophotometer was used to measure the absorbance of the solutions at 695 nm against a blank after they had cooled to room temperature. The samples were incubated under the same conditions as the blank solution, with the same quantity of reagent solution (300 µL) and the same solvent (300 mL) [54]. The number of ascorbic acid equivalents used to estimate the antioxidant activity is shown in the following mathematical equation:A = (c × V)/m
where A = total concentration of antioxidant compounds in plant extract compared to Ascorbic acid (mg/gm), c = Ascorbic acid content based on calibration curves (mg/mL), V = the volume of extract in mL, m = crude plant extract mass in grams.

#### 4.8.3. Determination of Total Flavonoids Content

A total of 1 mL plant extract (concentration 200 g/mL) or a reference solution of different concentrations were added to a test tube. A total of 3 mL methanol was added to the test tube. After that, 200 μL of a 10% aluminum chloride solution was added to the same test tube. After that, fill the test tube with 200 μL of 1M potassium acetate solution. At the end of the experiment, 5.6 mL of distilled water was added to the mixture. The mixture was incubated at room temperature for 30 min to complete the reaction. The absorbance of the solution at 415 nm was compared to that of a blank using a spectrum photometer. In the experiment, Methanol was used as a blank solution [55]. The total content of flavonoid compounds in plant methanol extracts in quercetin equivalents was calculated by the following formula equation:C_flavonoid_ = (c × V)/m
where C = total content of flavonoid contents (mg/gm) in plant extract in the equivalent to the quercetin, c = concentration of quercetin established from the calibration curve in mg/mL, V = volume of extract in mL, m = the mass of the crude extract of the plant in gm.

#### 4.8.4. Determination of Total Phenolic Contents

Total phenolic contents of the plant extracts were determined by following the established protocol mentioned by Ainsworth et al. [56]. Plant extracts (200 g/mL) or a standard solution at different proportion was used to fill the test tubes, each of which contained 5 mL. Folin–Ciocalteu reagent solution was also added to the test tubes (diluted 10-fold). A 7.5% sodium carbonate solution (4 mL) was added to the same test tube and well mixed. Standard solutions were incubated for 30 min at 20 °C, whereas extract solution was incubated for 1 h at 20 °C to complete the reaction. A spectrophotometer set to 765 nm was used to compare the solution’s absorbance to that of a blank reference sample. The solvent that was used to dissolve the plant extract was typically present in a blank solution. The following equation was used to determine the total phenolic component content of plant extracts in equivalents to Gallic acid:C_phenolic_ = (c × V)/m
where C = total content of phenolic contents (mg/gm) in the plant extract. c = the contents of Gallic acid established from the calibration curve (mg/mL), V = the volume of extract in mL, m = the weight of crude plant extract in gm.

### 4.9. In silico Analysis

#### 4.9.1. Molecular Docking: Protein Preparation

Urate oxidase (Uox) (PDB ID: 1R4U) [57], Glutathione reductase (PDB ID: 3GRS) [58], Hepatitis C Virus NS3/4A Protease (PDB ID: 3SU4) [59], and Human IgG Fc Domain (PDB ID: 4QGT) [60] have been derived from RCSB Protein Data Bank (https://www.rcsb.org/structure) (23 May 2022) in PDB format for the antioxidant and hepatoprotective activity, respectively. All water and heteroatoms have been removed from proteins using Discovery Studio 2020. To prepare proteins, the nonpolar hydrogens and the Gasteiger charge were retained at their default settings. UCSF Chimera was also used to reduce all proteins to their lowest energy state and process them for further analysis keeping them in the normal residues in AMBER ff14sB and other residues in Gasteiger mode [40].

#### 4.9.2. Molecular Docking: Ligand Preparation

The chemical structure of seven compounds of *Gynura nepalensis* entitled: 3-P-Coumaroylquinic Acid (PubChem CID: 9945785), 4,5-Dicaffeoylquinic Acid (PubChem CID: 6474309), Boscialin (PubChem CID: 6442487), Chlorogenic acid (PubChem CID: 1794427), Ethyl caffeate (PubChem CID: 5317238), Dibutyl phthalate (PubChem CID: 3026), Loliolide (PubChem CID: 100332), Medioresinol (PubChem CID: 181681), Propiconazole (PubChem CID: 43234) were retrieved from PubChem database (https://pubchem.ncbi.nlm.nih.gov/; accessed on 22 May 2022, Figure 5). Moreover, Ascorbic acid (PubChem CID: 54670067) and Betaine (PubChem CID: 247) have been taken as standard drugs and docked to compare and contrast the binding affinity of the selected compounds of *Gynura nepalensis*. The ligands have been downloaded in 2DSDF format and have been minimized using the PyRx AutoDock Vina in order to obtain the optimum feasible hit for these particular targets [12].

#### 4.9.3. Molecular Docking: Docking Analysis

The PyRx AutoDock Vina has been deployed for the docking of the selected protein-ligand complexes [24]. A semi-flexible docking mechanism was used for the docking investigation. With the help of PyRx AutoDock Vina, phytochemical and protein PDB files were minimized before being transformed to PDBQT. The protein’s stiffness and ligand’s versatility were preserved in this investigation. The ligand molecules have been given 10 degrees of flexibility. AutoDock defines the steps to be taken to transform the macromolecules [61].

## 5. Conclusions

According to the research, the methanol extract of *G. nepalensis* protected rats against paracetamol-induced liver failure in a dose-dependent manner. The hepatoprotective action may be attributed to its strong antioxidant properties since antioxidants protect liver tissue from degradation. Furthermore, the plant includes a variety of bioactive phytochemicals, and the interactions of these compounds with target proteins seem to be essential elements toward giving liver protection and may help to maximize the plant’s potential against liver diseases. As a result, in order to obtain comprehensive knowledge of the biological activity of the liver along with exact mechanism of actions, further advanced research on this plant is recommended.

## Figures and Tables

**Figure 1 molecules-27-03474-f001:**
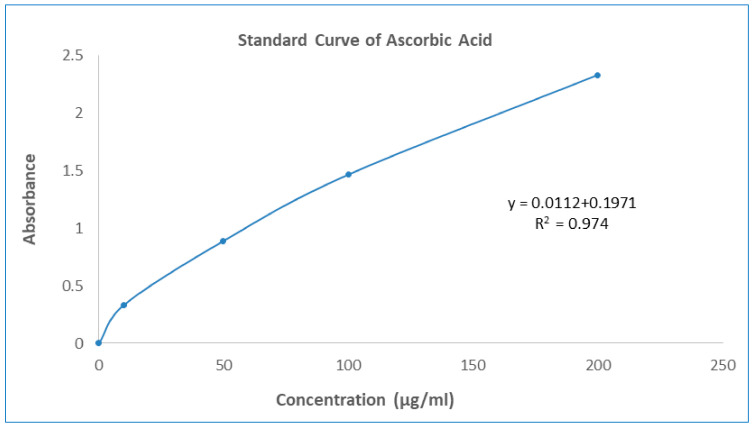
Standard curve of Ascorbic Acid.

**Figure 2 molecules-27-03474-f002:**
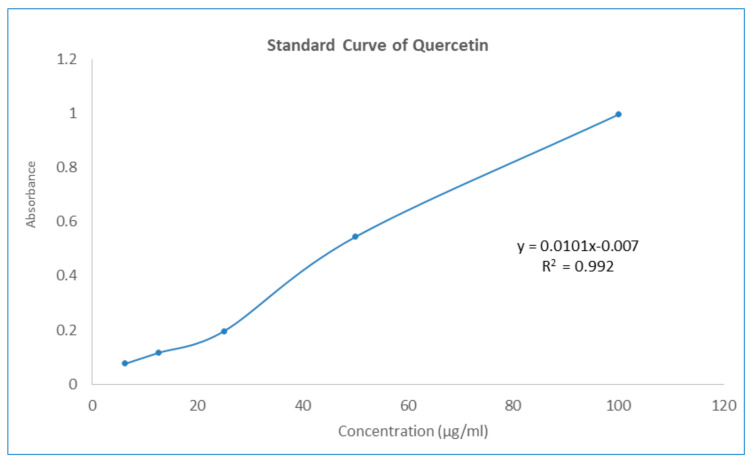
Standard curve of Quercetin.

**Figure 3 molecules-27-03474-f003:**
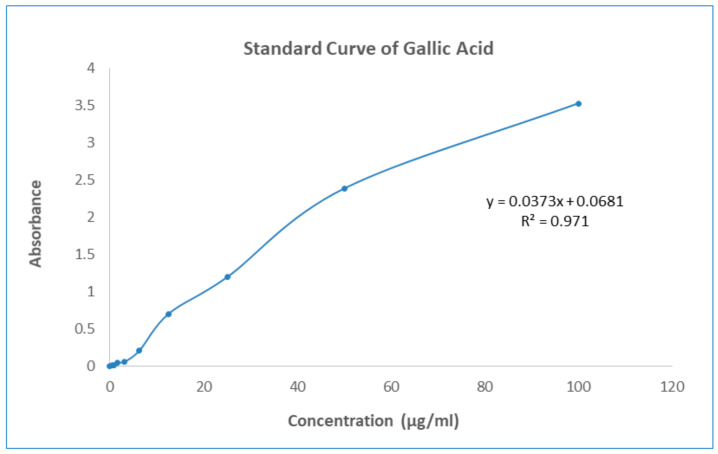
Standard curve of Gallic Acid.

**Figure 4 molecules-27-03474-f004:**
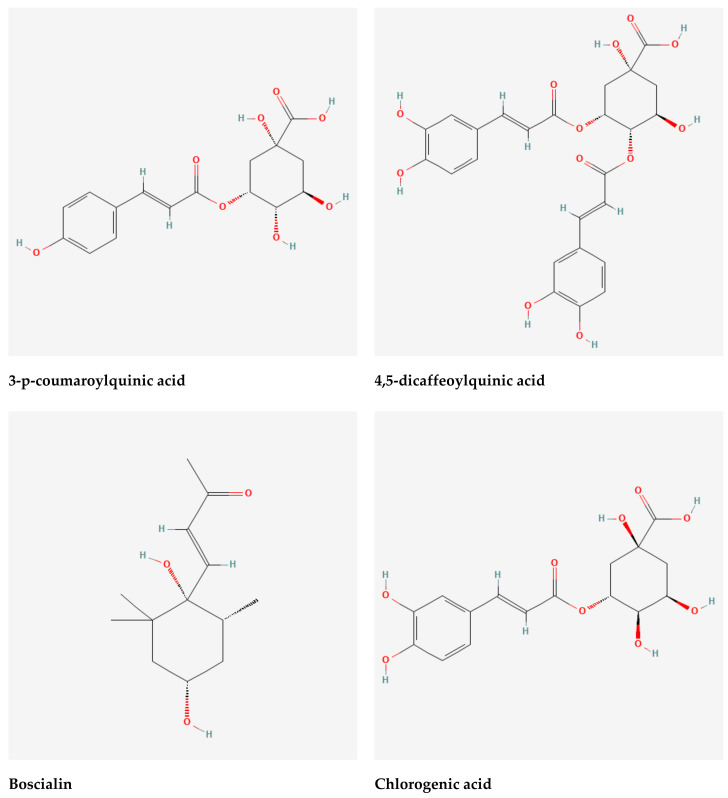
Selected phytochemicals (ligands) of the extracts of *Gynura nepalensis*.

**Figure 5 molecules-27-03474-f005:**
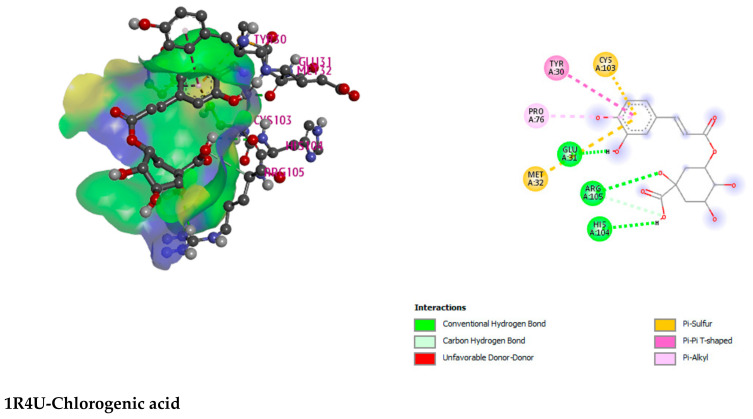
Illustration (2D and 3D) of the docking results of the best-ranked pose of key interactions in the binding pocket of Urate oxidase (Uox) (PDB ID: 1R4U) with Chlorogenic acid, Glutathione reductase (PDB ID: 3GRS) with 4,5-dicaffeoylquinic acid, Hepatitis C Virus NS3/4A Protease (PDB ID: 3SU4) with 4,5-dicaffeoylquinic acid, Human IgG Fc Domain (PDB ID: 4QGT) with 4,5-dicaffeoylquinic acid, respectively.

**Table 1 molecules-27-03474-t001:** Qualitative screening of GNME.

Group	Presence (+)/Absence (−)
Alkaloids	+
Carbohydrates Saponins	+
Tannins	+
Condensed Tannin	+
Terpenoids	+
Chlorogenic acid	+
Steroidal Glycosides	+
Anthocyanin	+
Flavonoids	+
Flavones	+
Phenols	+
Coumarins	+
Nitrogenous compounds	+
Quercetin	−
Triterpene	−
Coumarin	−

**Table 2 molecules-27-03474-t002:** Body and liver weight ratio of rats after the administration of samples.

Groups	Body Weight (gm)	Weight of Liver (gm)	Ratio of Bodyweight and Liver Weight
I: Water 10 mL/kg	184.8 ± 30.51	5.45 ± 0.37	0.030 ± 0.003
II: Water 10 mL/kg + Paracetamol 2 gm/kg	204.33 ± 57.79 ^#^	6.88 ± 1.52 ^#^	0.042 ± 0.002 ^#^
III: Paracetamol 2 gm/kg + GNME 100 mg/kg	170.25 ± 44.95 **	6.18 ± 1.38	0.035 ± 0.002 *
IV: Paracetamol 2 gm/kg + GNME 200 mg/kg	173.2 ± 39.69 **	6.16 ± 1.43	0.036 ± 0.004 **
V: Paracetamol 2 gm/kg + GNME 400 mg/kg	168.25 ± 34.30 **	5.97 ± 1.19 *	0.036 ± 0.001 **
VI: Paracetamol 2 gm/kg + Silymarin 100 mg/kg	159.8 ± 36.66 ***	5.27 ± 0.87 *	0.033 ± 0.002 ***

Values are presented as mean ± SEM; one-way analysis of variance (ANOVA) was followed by Dunnett’s test. * *p* < 0.05, ** *p* < 0.01 and *** *p* < 0.001 was considered as significant compared with the control, where ^#^ is designated as control.

**Table 3 molecules-27-03474-t003:** Effects of GNME on serum liver function parameters in different groups of experimental rats.

Groups	TB (mg/dL)	LP (nmol MDA/mg of Hb)	AST (U/L)	ALT (U/L)	ALP (U/L)	TP (mg/dL)
I: Water 10 mL/kg	0.27 ± 0.1	0.1 ± 0.01	253.2 ± 3.23	326.4 ± 6.59	259.2 ± 8.45	0.47 ± 0.01
II: Water 10 mL/kg + Paracetamol 2 gm/kg	1.32 ± 0.05 ^#^	0.19 ± 0.04 ^#^	271.6 ± 2.39 ^#^	357.2 ± 3.15 ^#^	283.4 ± 4.09 ^#^	0.54 ± 0.04 ^#^
III: Paracetamol 2 gm/kg + GNME 100 mg/kg	0.88 ± 0.16 ***	0.17 ± 0.03	196.20 ± 3.60 ***	216.80 ± 4.87 ***	234.6 ± 4.65 ***	0.46 ± 0.02 **
IV: Paracetamol 2 gm/kg + GNME 200 mg/kg	0.60 ± 0.09 **	0.15 ± 0.03	144.60 ± 7.86 ***	182.60 ± 4.48 ***	231.4 ± 3.00 *	0.42 ± 0.03 **
V: Paracetamol 2 gm/kg + GNME 400 mg/kg	0.34 ± 0.08 ***	0.12 ± 0.02 ***	114.60 ± 5.23 ***	142.20 ± 2.16 ***	191.6 ± 2.11 ***	0.45 ± 0.02 **
VI: Paracetamol 2 gm/kg + Silymarin 100 mg/kg	0.47 ± 0.06 ***	0.12 ± 0.02 ***	177.60 ± 2.78 ***	132.20 ± 5.64 ***	153.6 ± 5.52 ***	0.40 ± 0.03 *

One-way analysis of variance (ANOVA) was followed by Dunnett’s test (*n* = 5). * *p* < 0.05, ** *p* < 0.01 and *** *p* < 0.001 was considered as significant compared with the control, where ^#^ is designated as control. Total bilirubin (TB), lipid peroxidation (LP), alanine transaminase (ALT), aspartate aminotransferase (AST), alkaline phosphatase (ALP), and total protein (TP). Values are presented as mean ± SEM.

**Table 4 molecules-27-03474-t004:** Activity of Ascorbic Acid in DPPH scavenging assay.

Test Sample	Concentration (µg/mL)	Absorbance	%Inhibition	Line Equation	R^2^ Value	IC_50_ (µg/mL)
Ascorbic Acid	2.5	0.815 ± 0.81	5.63 ± 0.23	y = 4.7929x − 8.2238	0.9972	12.15
	5	0.723 ± 0.73	15.26 ± 0.81			
	10	0.529 ± 0.54	37.14 ± 1.45			
	20	0.119 ± 0.10	88.80 ± 2.61			
	40	0.033 ± 0.03	96.34 ± 0.17			
	80	0.026 ± 0.03	96.98 ± 0.00			

**Table 5 molecules-27-03474-t005:** Activity of GNME in DPPH scavenging assay.

Test Sample	Concentration (µg/mL)	Absorbance	%Inhibition	Line Equation	R^2^ Value	IC_50_ (µg/mL)
GNME	6.25	0.855 ± 0.86	0.46 ± 0.29	y = 0.2335x + 2.3806	0.9845	203.94
	12.5	0.832 ± 0.84	3.08 ± 0.35			
	25	0.804 ± 0.79	7.78 ± 1.10			
	50	0.741 ± 0.74	13.64 ± 0.35			
	100	0.617 ± 0.61	28.96 ± 0.58			
	200	0.341 ± 0.37	56.76 ± 3.66			
	400	0.072 ± 0.07	91.35 ± 0.29			
	800	0.065 ± 0.07	92.11 ± 0.35			

**Table 6 molecules-27-03474-t006:** Total antioxidant capacity, total flavonoid content, and total phenol content of GNME.

Test Sample	Absorbance 1	Absorbance 2	Average Absorbance	Total Antioxidant Capacity (mg Ascorbic Acid Equivalent/gm Extract)	Total Flavonoid Content (mg Quercetin Equivalent/gm Extract)	Total Phenol Content (mg Gallic Acid Equivalents/gm Extract)
GNME	1.035	1.031	1.033	74.63	102.97	25.87

**Table 7 molecules-27-03474-t007:** Docking scores of 3-P-Coumaroylquinic acid, 4,5-Dicaffeoylquinic acid, Boscialin, Chlorogenic acid, Ethyl caffeate, Dibutyl phthalate, Loliolide, Medioresinol, Propiconazole and standard drugs (Ascorbic acid/Betaine) with Urate oxidase (Uox), Glutathione reductase, Hepatitis C Virus NS3/4A Protease, and Human IgG Fc Domain in kcal/mol.

Docking Scores (Kcal/mol)					
Compounds	PubChem CID	Antioxidant		Hepato-Protective	
		1R4U	3GRS	3SU4	4QGT
3-P-Coumaroylquinic acid	9945785	−6.2	−8.2	−7.5	−6.1
4,5-Dicaffeoylquinic acid	6474309	−6.1	−10.1	−7.8	−6.5
Boscialin	6442487	−6.3	−5.8	−6.4	−4.6
Chlorogenic acid	1794427	−7.8	−8.1	−7.2	−5.8
Ethyl caffeate	5317238	−6.5	−6.3	−5.7	−4.7
Dibutyl phthalate	3026	−6.2	−6.6	−5.2	−4.4
Loliolide	100332	−6.5	−6.1	−6.1	−4.9
Medioresinol	181681	−6.3	−8.3	−5.4	−5.9
Propiconazole	43234	−7.5	−7.9	−5.8	−5.4
Standard drugs (Ascorbic acid/Betaine)	54670067/247	−4.3	−5.8	−3.6	−2.9

PDB: 1R4U = Urate oxidase (Uox), PDB: 3GRS = Glutathione reductase, PDB: 3SU4 = Hepatitis C Virus NS3/4A Protease and PDB: 4QGT = Human IgG Fc Domain; Bold Number = The Highest Binding Affinity.

## Data Availability

The data presented in this study are available on request from the corresponding author.

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
