# Peer review of "Chemico-Pharmacological Screening of the Methanol Extract of Gynura nepalensis D.C. Deciphered Promising Antioxidant and Hepatoprotective Potentials: Evidenced from in vitro, in vivo, and Computer-Aided Studies"

_molecules, 2022, doi:10.3390/molecules27113474_

Round 1
Reviewer 1 Report
1.From the conclusion of Table 1, plants contain a large number of non-volatile components. Why don't the author use LC/UV or LC/MS for analysis?
The GC/MS results in Table 2 seem meaningless.
2.What are the selection criteria for compounds in molecular docking experiments?
3.The size of the picture is too large.
4.The author measured the contents of total flavonoids and total phenols. The subtext is whether flavonoids or polyphenols have the effect of protecting the liver? What about other type components in Table 1?
Author Response
Dear Editor,
Thank you for your letter and the reviewer's comments concerning our manuscript entitled “Chemico-pharmacological screening of the methanol extract of Gynura nepalensis D.C. deciphered promising antioxidant and hepatoprotective potentials: Evidenced from in vitro, in vivo, and computer-aided studies’’(Manuscript ID: molecules-1732033). Those constructive comments were welcomed, reviewed, and interpreted by our research team to revise and improve the manuscript's quality. Given the highest importance to the commentaries, we have carefully studied the feedback from you and the reviewers and tried our best to revise the manuscript according to the instructions given by your experts. We have tried to revise the manuscript as per the commentaries and your journal's standards. We hope that our revised manuscript will fulfill the requirements and be accepted by your majesty for publishing the work in your reputed journal. The point-to-point response to the Editor and reviewers' comments have been attached below:
We are looking forward to getting good sounds soon from you!
Reviewer 1
- From the conclusion of Table 1, plants contain a large number of non-volatile components. Why don't the author use LC/UV or LC/MS for analysis?
The GC/MS results in Table 2 seem meaningless.
Reply 1: We are thanking reviewer 1 for his scholarly opinion. The plant Gynura nepalensis was subjected to GC-MS analysis technique but the findings showed some erroneous types of compounds like non volatile substances. However, based on previous researches utilizing HPLC, NMR and/or LC-MS techniques, several types of compunds including both volatile and non-volatile compounds have been identified from this plant. Thus, we have discarded the GC-MS data and focused on the previously reported phytochemicals to predict the propspective phytocompounds with antioxidant and hepatoprotective actions.
- What are the selection criteria for compounds in molecular docking experiments?
Reply 2: We have made an extensive literature search before conducting the study. There we have found several compounds from this plant confirmed by different types of analytical techniques including HPLC, NMR, LC-MS and so on. If we would focus on the compounds reperted by only fsingle type of analytical method, we may miss the responsible compounds which may be not traceable through that evaluation. Thus, we have selected candidates from different types of analytical techniques.
- The size of the picture is too large.
Reply 3: The large-sized pictures have been provided so that the production team will not face difficulties to find the pictures of adequate resolution. If the size is too large, the production team may adjust the size as per their requirement.
- The author measured the contents of total flavonoids and total phenols. The subtext is whether flavonoids or polyphenols have the effect of protecting the liver? What about other type components in Table 1?
Reply 4: We are thanking the reviewer for his scholarly opinion here. The role of different phytochemicals in hepato protection has been discussed in the discussion section.

Reviewer 2 Report
Authors should kindly attend to the minor corrections indicated in the reviewed manuscript.

Author Response
Dear Editor,
Thank you for your letter and the reviewer's comments concerning our manuscript entitled “Chemico-pharmacological screening of the methanol extract of Gynura nepalensis D.C. deciphered promising antioxidant and hepatoprotective potentials: Evidenced from in vitro, in vivo, and computer-aided studies’’(Manuscript ID: molecules-1732033). Those constructive comments were welcomed, reviewed, and interpreted by our research team to revise and improve the manuscript's quality. Given the highest importance to the commentaries, we have carefully studied the feedback from you and the reviewers and tried our best to revise the manuscript according to the instructions given by your experts. We have tried to revise the manuscript as per the commentaries and your journal's standards. We hope that our revised manuscript will fulfill the requirements and be accepted by your majesty for publishing the work in your reputed journal. The point-to-point response to the Editor and reviewers' comments have been attached below:
We are looking forward to getting good sounds soon from you!
Reviewer 2
Reply: We are thanking reviewer 2 very much for his/her scholarly opinion to improve the overall quality of the manuscript. We have incorporated all suggestions in our revised manuscript.

Reviewer 3 Report
Based on an extensive analysis of the presented data, I am afraid that I have to reject this manuscript. The reasons are:
- The identification of compounds by GC-MS is not convincing. The author identified several non-volatile compounds such as flavonoid and phenolic glycosides in GC-MS without providing a chromatogram, and the mass spectrum of respective compounds is not acceptable.
- The calibration curves for ascorbic acid (TAC), quercetin (TFC), and gallic acid (TPC) are out of the acceptable range, with R2 generally less than 0.999, indicating the in vitro antioxidant data is not reliable.
- The docking proteins have no direct correlation with the in vivo or in vitro assay. The docking finding is too superficial.
Author Response
Dear Editor,
Thank you for your letter and the reviewer's comments concerning our manuscript entitled “Chemico-pharmacological screening of the methanol extract of Gynura nepalensis D.C. deciphered promising antioxidant and hepatoprotective potentials: Evidenced from in vitro, in vivo, and computer-aided studies’’(Manuscript ID: molecules-1732033). Those constructive comments were welcomed, reviewed, and interpreted by our research team to revise and improve the manuscript's quality. Given the highest importance to the commentaries, we have carefully studied the feedback from you and the reviewers and tried our best to revise the manuscript according to the instructions given by your experts. We have tried to revise the manuscript as per the commentaries and your journal's standards. We hope that our revised manuscript will fulfill the requirements and be accepted by your majesty for publishing the work in your reputed journal. The point-to-point response to the Editor and reviewers' comments have been attached below:
We are looking forward to getting good sounds soon from you!
Reviewer 3
- The identification of compounds by GC-MS is not convincing. The author identified several non-volatile compounds such as flavonoid and phenolic glycosides in GC-MS without providing a chromatogram, and the mass spectrum of respective compounds is not acceptable.
Reply 1: We are thanking reviewer 3 for his scholarly opinion. The plant Gynura nepalensis was subjected to GC-MS analysis technique but the findings showed some erroneous types of compounds like non volatile substances. However, based on previous researches utilizing HPLC, NMR and/or LC-MS techniques, several types of compunds including both volatile and non-volatile compounds have been identified from this plant. Thus, we have discarded the GC-MS data and focused on the previously reported phytochemicals to predict the propspective phytocompounds with antioxidant and hepatoprotective actions.
- The calibration curves for ascorbic acid (TAC), quercetin (TFC), and gallic acid (TPC) are out of the acceptable range, with R2 generally less than 0.999, indicating the in vitro antioxidant data is not reliable.
Reply 2: In biochemical analysis, sometimes it has become difficult to find the R2 value of exactly 0.999. We have tried a lot to be more close to the value 0.999 and thus, our obtained value is almost close to 0.999 which may demonstrate the good standard curve for samples. However, we have redone the standard curve of Gallic acid (R2 value 0.971) with more precautions to obtain a better result.
- The docking proteins have no direct correlation with the in vivo or in vitro assay. The docking finding is too superficial.
Reply 3: The targeted docking proteins have been selected based on literature searches. According to previous reports, the concerned pharmacological actions are attributed to these proteins. Thus, we have docked these proteins to bridge the knowledge between our findings with previous reports.

Round 2
Reviewer 1 Report
The author has completed relevant modifications. I suggest accepting the manuscript
Author Response
Dear Editor,
Thank you for your letter and the reviewer's comments concerning our manuscript entitled “Chemico-pharmacological screening of the methanol extract of Gynura nepalensis D.C. deciphered promising antioxidant and hepatoprotective potentials: Evidenced from in vitro, in vivo, and computer-aided studies’’(Manuscript ID: molecules-1732033). Those constructive comments were welcomed, reviewed, and interpreted by our research team to revise and improve the manuscript's quality. Given the highest importance to the commentaries, we have carefully studied the feedback from you and the reviewers and tried our best to revise the manuscript according to the instructions given by your experts. We have tried to revise the manuscript as per the commentaries and your journal's standards. We hope that our revised manuscript will fulfill the requirements and be accepted by your majesty for publishing the work in your reputed journal. The point-to-point response to the Editor and reviewers' comments have been attached below:
We are looking forward to getting good sounds soon from you!
Reviewer 2
Reviewer’s comment: The author has completed relevant modifications. I suggest accepting the manuscript.
Reply: We are thanking reviewer 2 very much for his/her scholarly opinion.

Reviewer 3 Report
The authors gave a reasonable response to the arisen problems. I have some minor suggestions for the authors to improve the quality of the manuscript:
1. I agree that the authors removed the GC-MS profile and only performed the docking analysis based on the reported compounds found in the literature. For section 2.5, please add a sentence that the docked or selected compounds are the major phytochemicals in the plant with the references.
2. For the calibration curves, I noticed that the authors drew a random curve in Figures 1 to 3. Please change to a linear curve.
Author Response
Dear Editor,
Thank you for your letter and the reviewer's comments concerning our manuscript entitled “Chemico-pharmacological screening of the methanol extract of Gynura nepalensis D.C. deciphered promising antioxidant and hepatoprotective potentials: Evidenced from in vitro, in vivo, and computer-aided studies’’(Manuscript ID: molecules-1732033). Those constructive comments were welcomed, reviewed, and interpreted by our research team to revise and improve the manuscript's quality. Given the highest importance to the commentaries, we have carefully studied the feedback from you and the reviewers and tried our best to revise the manuscript according to the instructions given by your experts. We have tried to revise the manuscript as per the commentaries and your journal's standards. We hope that our revised manuscript will fulfill the requirements and be accepted by your majesty for publishing the work in your reputed journal. The point-to-point responses to the reviewers’ comments have been attached below:
We are looking forward to getting good sounds soon from you!
Reviewer 3
Reviewer’s comment 1: I agree that the authors removed the GC-MS profile and only performed the docking analysis based on the reported compounds found in the literature. For section 2.5, please add a sentence that the docked or selected compounds are the major phytochemicals in the plant with the references.
Authors’ reply 1: We are thanking reviewer 2 for his/her scholarly comment to increase the overall quality of the manuscript. We have included a sentence in section 2.5 along with an appropriate reference (reference 18) to validate the claim.
Reviewer’s comment 2: For the calibration curves, I noticed that the authors drew a random curve in Figures 1 to 3. Please change to a linear curve.
Authors’ reply 2: We are thanking reviewer 2 for his/her scholarly comment to increase the overall quality of the manuscript. The values of each standard curve are displayed here. We avoided the values of the curve to avoid similar types of data. But upon the suggestion of the respected reviewer, we are tabulating the data here.
|
Gallic Acid |
|||||
|
SL. No. |
Conc. of the Standard (µg / ml) |
Absorbance |
Regression line |
R2 |
|
|
1 |
100 |
3.522 |
y = 0.0373x + 0.0681 |
0.971 |
|
|
2 |
50 |
2.381 |
|||
|
3 |
25 |
1.192 |
|||
|
4 |
12.5 |
0.702 |
|||
|
5 |
6.25 |
0.207 |
|||
|
6 |
3.125 |
0.059 |
|||
|
7 |
1.5625 |
0.044 |
|||
|
8 |
0.78125 |
0.015 |
|||
|
9 |
0.3906 |
0.007 |
|||
|
10 |
0 |
0.004 |
|||
|
Ascorbic Acid |
|||||
|
SL. No. |
Conc. of the Standard (µg / ml) |
Absorbance |
Regression line |
R2 |
|
|
1 |
0 |
0 |
y = 0.0112x + 0.1971 |
0.9736 |
|
|
2 |
10 |
0.334 |
|||
|
3 |
50 |
0.89 |
|||
|
4 |
100 |
1.464 |
|||
|
5 |
200 |
2.328 |
|||
|
Quercetin |
|||||
|
SL. No. |
Conc. of the Standard (µg / ml) |
Absorbance |
Regression line |
R2 |
|
|
1 |
100 |
0.995 |
y = 0.0101x - .007 |
0.9916 |
|
|
2 |
50 |
0.544 |
|||
|
3 |
25 |
0.196 |
|||
|
4 |
12.5 |
0.116 |
|||
|
5 |
6.25 |
0.076 |
|||

This manuscript is a resubmission of an earlier submission. The following is a list of the peer review reports and author responses from that submission.